# Overexpression of *CDC20* Confer a Poorer Prognosis in Bladder Cancer Identified by Gene Co-Expression Network Analysis

**DOI:** 10.3390/diagnostics15233016

**Published:** 2025-11-27

**Authors:** Xuejian Yang, Yunjie Guo, Lanyu Wang, Zengli Miao, Xiaojie Lu, Jun Ruan, Wei Tian

**Affiliations:** 1Department of Urology, Suqian First Hospital, Suqian 223800, China; yangxuejiansq@163.com; 2Department of Urology, The Affiliated Wuxi People’s Hospital of Nanjing Medical University, Wuxi People’s Hospital, Wuxi Medical Center, Nanjing Medical University, Wuxi 214000, China; gyjkd35@163.com; 3The First Affiliated Hospital of Nanchang University, Nanchang 330000, China; wanglanyuage@163.com; 4Department of Urology, Jiangnan University Medical Center (The Affiliated Wuxi No. 2 People’s Hospital), Wuxi 214000, China; 5Department of Neurosurgery, Jiangnan University Medical Center (The Affiliated Wuxi No. 2 People’s Hospital), Wuxi 214000, China; 9862023218@jiangnan.edu.cn (Z.M.); luxiaojiewuxi@163.com (X.L.); 6Wuxi School of Medicine, Jiangnan University, Wuxi 214000, China; 7Wuxi Neurosurgical Institute, Wuxi 214000, China; 8Department of Neurosurgery, General Hospital of Tianjin Medical University, Tianjin Medical University, Tianjin 300000, China

**Keywords:** bioinformatics analysis, *CDC20*, biomarker, cancer progression, bladder cancer

## Abstract

**Background/Objectives**: Bladder cancer (BCa) ranks as the tenth most prevalent malignancy worldwide, characterized by high morbidity and mortality rates. Despite advancements in understanding its pathogenesis, the identification of robust prognostic biomarkers remains critical for improving clinical outcomes. This study aims to identify and validate novel prognostic markers for BCa through integrated bioinformatics and experimental approaches. **Methods**: Gene expression data and clinical information were obtained from the GEO (GSE13507) and TCGA databases. Differential gene expression analysis and weighted gene co-expression network analysis (WGCNA) were employed to identify overlapping genes. Functional enrichment analysis was performed to explore biological functions, followed by protein–protein interaction (PPI) network construction and survival analysis. Key candidate genes were screened using the CytoHubba plugin in Cytoscape. *CDC20* expression was validated through RT-qPCR, and its functional role in BCa cells was assessed in vitro. **Results**: Eight candidate hub genes (*TROAP*, *TPX2*, *TOP2A*, *KIF2C*, *AURKA*, *CDC20*, *PRC1*, and *AURKB*) were identified. Survival analysis revealed that high *CDC20* expression was significantly associated with decreased overall survival in BCa patients. Mechanistic investigations demonstrated that *CDC20* promotes tumor invasion and growth by modulating mitosis and cell cycle progression, while also influencing the tumor microenvironment through immune cell regulation. Experimental validation confirmed the tumor-promoting role of *CDC20* in BCa cells. **Conclusions**: This study identifies *CDC20* as a key prognostic biomarker for bladder cancer, providing novel insights for early diagnosis, clinical treatment, and prognosis assessment. The findings highlight the potential of *CDC20* as a therapeutic target and underscore the value of integrated bioinformatics and experimental validation in biomarker discovery.

## 1. Introduction

Bladder cancer (BCa) ranks as the tenth most common cancer worldwide and poses a significant burden in terms of morbidity and mortality [1]. Non-muscle-invasive bladder cancer (NMIBC) accounts for approximately 75% of all bladder cancer cases, but the risk of recurrence and progression to muscle-invasive bladder cancer (MIBC) varies [2]. The remaining 25% are MIBC, for which radical cystectomy combined with pelvic lymph node dissection is the current standard of care in localized cases. However, even with salvage treatment, the prognosis of patients is generally poor [3,4]. Current clinical tools are still limited in predicting pathological stages and response to immunotherapy [5,6]. Therefore, there is an urgent need to study the molecular pathogenesis of bladder cancer and to identify biomarkers that are closely related to its diagnosis, progression, and prognosis. Interrogation of bladder tumor tissue for molecular biomarkers—such as *FGFR3*, *VEGF-C*, *GATA3*, *FOXA1*, and *P53*—provides a foundation for predicting recurrence risk and therapeutic response. This approach, complemented by traditional histopathology, represents an advancement toward precision oncology in bladder cancer care [7].

During the tumor progression, tumor cells engage in continuous interactions with neighboring stromal cells, resulting in the formation of the tumor microenvironment (TME) [8]. The TME plays a pivotal role in creating a conducive environment for tumorigenesis, proliferation, invasion, and metastasis [9]. Tumor-infiltrating immune cells (TIICs) have emerged as key regulators of cancer cell behavior within the TME. Extensive research has elucidated the contributions of various TIICs, including tumor-associated macrophages, dendritic cells, neutrophils, mast cells, natural killer cells, and lymphocytes, to the development of distinct cancer hallmarks. Functionally plastic and heterogeneous tumor-infiltrating immune cells (TIICs) can exert opposing effects on tumor progression, yet they potentiate immunotherapy and suppress recurrence through direct tumor targeting, immune memory formation, regulation of immune subsets, and blockade of inhibitory signals [10]. These findings have laid the foundation for successful immunotherapy approaches [11]. Conversely, alterations in the composition of immune cells within the TME have been implicated in tumor progression and immune response. Therefore, it is crucial to identify and characterize genes associated with TIICs [12]. Such studies further enhance our understanding of immune cell infiltration in bladder cancer, help decipher the TME landscape, and improve the treatment and prognosis of BCa.

In this study, we employed weighted gene co-expression network analysis (WGCNA) and conducted differential gene expression analysis using BCa mRNA expression data obtained from the TCGA and GEO databases. The aim was to identify gene modules that are highly associated with BCa [13,14]. To gain insights into the mechanisms underlying BCa development, we further investigated functional enrichments and protein–protein interaction (PPIs) through survival analysis. Subsequently, after a series of screening and validation experiments, we identified the *CDC20* gene as a promoter of BCa progression with prognostic implications.

Therefore, this study used WGCNA to identify novel prognostic biomarkers and validate their expression patterns in bladder uroepithelial carcinoma. These findings could help to provide a new direction for therapeutic intervention in BCa in the field of precision medicine.

## 2. Materials and Methods

### 2.1. Acquisition of Raw Data

Bladder cancer gene expression data were retrieved from two complementary public repositories: The Cancer Genome Atlas (TCGA) platform (https://portal.gdc.cancer.gov/, accessed on 21 November 2025) and the Gene Expression Omnibus (GEO) database (http://www.ncbi.nlm.nih.gov/geo/, accessed on 21 November 2025). Initial data acquisition from TCGA was performed using the TCGAbiolinks R package (v2.28.0), yielding a cohort comprising 414 malignant specimens and 19 normal control tissues. Following quality control measures, which involved the exclusion of 6 duplicate entries and 2 cases with incomplete survival documentation, 406 tumor samples with fully annotated clinical records were retained for subsequent investigation. Gene filtration was conducted using a counts-per-million (CPM) threshold ≥ 1, followed by normalization through the RPKM algorithm (reads per kilobase per million mapped reads) implemented in the EdgeR package (v3.42.0). This processing pipeline identified 14,829 qualified genes with valid RPKM measurements for further examination.

Parallel analysis incorporated the GSE13507 dataset from GEO, accessed via the GEOquery R package (v2.68.0), containing 188 neoplastic and 68 normal bladder tissue specimens. Probe-to-gene symbol conversion was executed using platform-specific annotation files, with expression values calculated as the median intensity across all corresponding probes. This procedure generated a curated set of 24,343 uniquely annotated genes. To facilitate prognostic validation of candidate hub genes, an independent verification cohort was established using 224 tumor samples with complete survival metadata from the GSE32894 collection, originally containing 308 malignant specimens.

### 2.2. Identifying a Co-Expression Module by WGCNA

WGCNA represents a systematic methodology for identifying coordinated expression patterns across multiple biological samples. This approach clusters genes demonstrating similar expression profiles and evaluates their correlations with specific phenotypic traits. In the present study, we constructed co-expression networks using transcriptomic data from the TCGA-BLCA cohort and the GSE13507 dataset through the WGCNA R package (v1.72-5).

The analytical pipeline initiated with the computation of a Pearson’s correlation matrix to determine pairwise gene expression correlations. This was subsequently converted into a weighted adjacency matrix using the power transformation aij = |Sij|^^β^, where aij denotes the adjacency between genes i and j, Sij represents their Pearson’s correlation coefficient, and β indicates the soft threshold parameter. To approximate scale-free topology, we applied soft thresholds of β = 3 for TCGA-BLCA and β = 11 for GSE13507. The adjacency matrices were then transformed into topological overlap matrices (TOM), with corresponding dissimilarity measures (1-TOM) calculated for downstream analysis.

Hierarchical clustering of the 1-TOM enabled the identification of discrete gene modules characterized by synchronized expression patterns. The final network incorporated 14,513 genes from TCGA-BLCA and 8889 genes from GSE13507, resulting in an integrated dataset of 17,058 unique genes for subsequent module-trait relationship analysis.

### 2.3. Differential Expression Analysis and Interaction with Modules of Interest

Differential gene expression analysis between bladder carcinoma tissues and normal controls was conducted using the LIMMA package (v3.56.0), applied independently to both the TCGA-BLCA and GSE13507 datasets. Transcripts demonstrating expression changes with |logFC| ≥ 1.0 and statistical significance (*p* < 0.05) were classified as differentially expressed genes (DEGs). The ggplot2 R package (v3.4.0) facilitated visualization of these DEGs through volcano plots, highlighting expression patterns in both cohorts.

To pinpoint potential prognostic biomarkers, we identified intersecting genes between the DEG sets and co-expression network modules. The overlapping gene candidates from these analyses were graphically represented using the “VennDiagram” package (v1.7.3), enabling clear visualization of shared and unique transcriptional signatures across different analytical approaches.

### 2.4. Function Enrichment Analysis

The clusterProfiler R package (v4.8.0) provides comprehensive analytical tools for Gene Ontology (GO) and Kyoto Encyclopedia of Genes and Genomes (KEGG) enrichment investigations. GO annotation systematically characterizes gene functions through three complementary domains: biological processes (BP), describing coordinated cellular activities; cellular components (CC), defining subcellular localization; and molecular functions (MF), elucidating biochemical activities at the molecular level. For all enrichment analyses, terms achieving an adjusted *p*-value threshold of <0.05 were considered statistically significant.

### 2.5. PPI Network and Hub Gene Identification

To establish a protein–protein interaction (PPI) network, we employed the STRING database (https://string-db.org/, accessed on 21 November 2025) as our primary search tool. This platform enabled systematic interrogation of documented gene interactions, with a confidence threshold set at ≥0.4 to filter for biologically significant associations. All unconnected genes were systematically excluded from subsequent analysis. The refined network was then imported into Cytoscape (v3.8.2) for advanced visualization and topological examination.

For identification of pivotal regulators within the co-expression network, we implemented the maximal clique centrality (MCC) algorithm through the CytoHubba plugin. This approach has been previously validated as particularly effective for detecting central nodes in biological networks. Based on computed MCC scores, the ten genes exhibiting the highest centrality values were designated as hub genes for subsequent functional characterization.

### 2.6. Verification of the Prognostic Value of Hub Genes

To evaluate the prognostic relevance of identified hub genes, we conducted Kaplan–Meier survival analysis through the R survival package utilizing TCGA-BLCA datasets. This investigation specifically examined the correlation between hub gene expression profiles and overall survival (OS) outcomes. The cohort was restricted to patients with complete follow-up documentation, who were subsequently stratified into high- and low-expression subgroups according to median expression values of respective hub genes. Statistical significance of survival disparities was determined using the log-rank test, with a threshold of *p* < 0.05 indicating prognostic relevance for subsequent validation.

### 2.7. Verification of the Expressions of Hub Genes with Prognostic Value and Gene Set Enrichment Analysis

To further validate the expression patterns of prognostic hub genes, we performed comparative analysis between bladder carcinoma tissues and normal controls using the GEPIA online platform (http://gepia.cancer-pku.cn/, accessed on 21 November 2025). For functional annotation, we acquired the Hallmark gene sets from the Molecular Signatures Database (MSigDB) as reference collections. Gene set enrichment analysis (GSEA) was subsequently conducted using GSEA 3.0 software (Broad Institute, Cambridge, MA, USA), with significance determined by a false discovery rate (FDR) q-value threshold of <0.05 across all transcriptional profiles in the TCGA-BLCA cohort.

### 2.8. Tumor-Infiltrating Immune Cells

The CIBERSORT algorithm was employed to characterize the composition of tumor-infiltrating immune cells across all malignant specimens. Following initial quantification, we applied stringent quality control measures, retaining only 201 tumor samples demonstrating statistically significant deconvolution results (*p* < 0.05) for subsequent immune profiling analyses.

### 2.9. Cell Culture

The human bladder cancer cell lines EJ and T24, together with the immortalized human urothelial cell line SV-HUC-1, were acquired from the Cell Bank of Shanghai Institutes for Biological Sciences, Chinese Academy of Sciences (Shanghai, China). All cell lines were maintained in Roswell Park Memorial Institute (RPMI) 1640 basal medium (Scientific Cells, San Diego, CA, USA) supplemented with 10% fetal bovine serum and 100 U/mL penicillin-streptomycin antibiotic mixture (Gibco, Grand Island, NY, USA), under standard culture conditions. Information of regents is available in Appendix A.

### 2.10. siRNA Transfection

The *CDC20*-targeting siRNA (si-*CDC20*) and negative control siRNA (si-NC) were designed and synthesized by Sangon Bioengineering Co., Ltd. (Shanghai, China). For transfection experiments, EJ and T24 cells were plated in 6-well plates and allowed to reach 50% confluency. Following the manufacturer’s protocol for Lipofectamine 2000 (Invitrogen, Thermo Fisher Scientific, Carlsbad, CA, USA), we prepared transfection complexes containing either si-*CDC20* or si-NC at a final concentration of 50 nM. After 6 h of exposure, the transfection medium was replaced with complete growth medium, and cells were maintained for an additional 24 h before subsequent analysis. The specific siRNA sequences utilized were: si-*CDC20* sense (5′-CGGGAACUGUUAACCAAAUUATT-3′) and antisense (5′-UAAUUUGGUUAACAGUUCCCGTT-3′); negative control sense (5′-UUCUCCGAACGUGUCACGUTT-3′) and antisense (5′-ACGUGACACGUUCGGAGAATT-3′). Information of regents is available in Appendix A.

### 2.11. RNA Extraction and Real-Time Quantitative Polymerase Chain Reaction

To evaluate *CDC20* expression patterns, total RNA was isolated from EJ, T24, and SV-HUC-1 cell lines using TRIzol reagent (Invitrogen, USA). Complementary DNA was synthesized with the iScript cDNA Synthesis Kit (Bio-Rad, Hercules, CA, USA), followed by real-time quantitative PCR amplification on a LightCycler^®^ 480 platform (Roche, Mannheim, Germany). Relative quantification of target gene expression was calculated using the 2^−ΔΔCt^ method, with β-actin serving as the endogenous normalization control. Information of regents is available in Appendix A. Raw data for PCR is available in Appendix A.

### 2.12. Cell Counting Kit-8 Assay

Following transfection with either si-NC or si-*CDC20*, EJ and T24 cells were plated in 96-well plates at a density of 2 × 10^3^ cells per well, with triplicate wells per experimental condition. The day of complete cell adhesion was designated as day 0. Cellular proliferation was assessed at 24-h intervals over four consecutive days using the Cell Counting Kit-8 (CCK-8) Kit (Vazyme, Nanjing, China). According to the manufacturer’s protocol, cells were incubated with CCK-8 solution (culture medium: CCK-8 reagent = 10:1) for 2 h, after which optical density (OD) values were measured at 450 nm. The entire experimental procedure was independently replicated three times to ensure statistical reliability. Information of regents is available in Appendix A.

### 2.13. Transwell Assays

For invasion capability assessment, transfected EJ and T24 cells (20,000 cells per chamber) were seeded into Matrigel-coated Transwell (Corning, Corning, NY, USA) inserts in 200 µL of serum-free RPMI-1640 medium. The lower chambers were filled with 600 µL of complete medium containing 10% FBS as a chemoattractant. Following a 48-h incubation, non-invading cells on the upper membrane surface were mechanically removed. The migrated cells on the lower surface were fixed with 4% neutral buffered formalin for 30 min and stained with 1% crystal violet solution for quantitative analysis. Information of regents is available in Appendix A.

### 2.14. Statistical Analysis

All statistical analyses were conducted using SPSS 18.0 (Chicago, IL, USA), GraphPad Prism 7.0 (La Jolla, CA, USA), and R software (v4.3.2) with essential packages including “Limma v3.56.0”, “TCGAbiolinks v2.28.0”, “ComplexHeatmap 2.16.0”, “ClusterProfile v4.8.0”, and “ggplot2 v3.4.0”. Continuous variables are presented as mean ± standard deviation (SD). Group comparisons were performed using appropriate statistical methods: the two-tailed Student’s *t*-test for normally distributed data, the Mann–Whitney U test for non-parametric comparisons, and the Chi-square test for categorical variables. A *p*-value below 0.05 was considered statistically significant for all analyses.

## 3. Results

### 3.1. Gene Intersection Between Differentially Expressed Genes and Co-Expression Modules

Based on the specified cutoff criteria (|LogFC| > 1, FDR < 0.05), the LIMMA package identified 916 and 464 Differentially Expressed Genes (DEGs) in the TCGA and GSE13507 datasets, respectively (Figure 1A–D). Subsequently, genes that were either upregulated or downregulated in both datasets were selected, resulting in 35 co-upregulated and 207 co-downregulated genes (Figure 1E,F).

### 3.2. Construction of Weighted Co-Expression Network and Identification of Key Modules

The WGCNA package was utilized to construct gene co-expression networks employing the TCGA-BLCA and GSE13507 datasets. The networks produced were partitioned into distinct modules, each denoted by a specific color. In this investigation, a total of 12 modules were delineated in TCGA-BLCA (Figure 2C), while 11 modules were identified in GSE 13507 (Figure 2I). To evaluate the relationship between each module and two clinical features (cancer and normal), heat maps illustrating module-feature associations were generated. As shown in Figure 2D,J, the purple module in TCGA-BLCA and the brown module in GSE13507 display the strongest correlation with bladder cancer tissue. Within the TCGA dataset, 4080 and 3357 co-expressed genes were identified in the purple and brown modules, respectively, while in the GSE 13507 dataset, 4080 and 3357 co-expressed genes were found in the blue-green and brown modules, respectively. Additionally, 2340 cell cycle-related genes were included in the study, from which 39 overlapping genes were extracted and utilized to validate the genes in the co-expression module (Figure 3A).

### 3.3. Gene Set Enrichment Analysis

To elucidate the potential biological functions and pathway relevance of the gene sets, we conducted functional enrichment analysis on the 39 overlapping genes. Gene Ontology (GO) analysis revealed enrichment in mitotic processes, spindle organization, and microtubule binding, suggesting their involvement in cell division and microtubule-related functions. Additionally, KEGG pathway analysis highlighted significant enrichment in the cell cycle pathway, indicating the pivotal role of these genes in regulating various phases of the cell cycle. Interestingly, enrichment in the T-cell leukemia virus (HTLV-1) infection pathway was also observed, suggesting a potential association with processes related to viral infection (Figure 3B).

### 3.4. Construction of PPI Network and Validation of CDC20 Gene Identification

The PPI network among the overlapping genes was constructed using the STRING database, excluding those not associated with other genes. The visualization of the PPI network was depicted in Figure 3C–F through Cytoscape software (v3.8.2). Utilizing the Maximum Clique Centrality (MCC) algorithm of the CytoHubba plugin, eight genes—*TROAP*, *TPX2*, *TOP2A*, *KIF2C*, *AURKA*, *CDC20*, *PRC1*, and *AURKB*—were identified as candidate central genes with the highest association (Figure 3G).

Subsequently, we assessed the prognostic value of these eight genes in TCGA using survival-related data. Overall Survival analysis of the eight hub genes was conducted using univariate Cox regression analysis. The analysis revealed that among the eight hub genes, only *CDC20* expression significantly correlated with OS in BCa patients (*p* < 0.05; Figure 3H), indicating that high *CDC20* expression is associated with poor prognosis in BCa patients. We further examined *CDC20* expression in paired and unpaired tissues from the TCGA dataset and analyzed its correlation with clinical characteristics (Table 1 and Table 2) (Detailed description is available in Appendix A). The multivariate analysis substantiates tumor grade as an independent prognostic factor. Furthermore, the significant association between elevated *CDC20* expression and aggressive pathological characteristics strongly supports its dual potential as a prognostic biomarker and therapeutic target. Importantly, we demonstrated that *CDC20* is upregulated in bladder cancer tissues and cell lines, and this elevated expression predicts poorer survival outcomes (Figure 4). Collectively, these findings position *CDC20* as a key driver of tumor aggressiveness and an indicator of unfavorable prognosis.

### 3.5. Functional Enrichment of CDC20 and Validation of GSEA

The protein–protein interaction network analysis of *CDC20* interactions using the “ComPPI” website provided insights into the molecular interactions of *CDC20* with other proteins involved in bladder cancer (Figure 5). The GO enrichment analysis of *CDC20*-associated genes revealed significant enrichment in mitotic processes, chromosome condensation, histone kinase activity, spindle, and microtubule binding. These findings suggest the involvement of *CDC20* and its associated genes in regulating various aspects of cell division and chromosomal organization. Among the KEGG pathway analysis, the *CDC20*-associated genes were predominantly enriched in the cell cycle pathway, indicating their role in the regulation of cell cycle progression. Additionally, the progesterone-mediated oocyte maturation pathway was also enriched, suggesting a potential link between *CDC20* and reproductive processes.

GSEA analysis was performed on samples with high and low *CDC20* expression levels. The results using the Hallmark pathway database indicated that the genes in the high *CDC20* expression group were significantly enriched in several pathways, including E2F targets, G2/M checkpoints, allograft rejection, Myc targets v1, Interferon Gamma (IFN-γ) responses, inflammatory responses, mTORC1 signaling, the mitotic spindle, and complement-related pathways. These findings suggest that high *CDC20* expression may be associated with the activation of these pathways and potentially contribute to oncogenic processes and immune responses in the tumor microenvironment (TME).

### 3.6. Correlation Between CDC20 and Tumor Immune Microenvironment

To further confirm the correlation between *CDC20* expression and the immune microenvironment, the proportion of tumor-infiltrating immune subpopulations was analyzed using the CIBERSORT algorithm, and 24 types of immune cell profiles were constructed in TCGA-BLCA samples (Figure 6). The results of the difference and correlation analysis showed that *CDC20* expression was positively correlated with the number of Th2 and Th1 cells and inversely correlated with the number of NK cells and mast cells. These results further support the effect of *CDC20* levels on the immune activity of TME.

### 3.7. Expression and Functional Characterization of CDC20

The impact of *CDC20* downregulation on cell proliferation was assessed using the CCK-8 assay. The findings revealed a significant inhibition in the proliferation of both bladder cancer cell lines upon reduction in CDC20 expression (Figure 7A). To further elucidate the role of *CDC20* in bladder cancer cell proliferation, clone formation assays were conducted. The results illustrated a reduction in the number of clones formed by bladder cancer cells following *CDC20* knockdown, indicating a suppressive effect on cell proliferation (Figure 7C,D). Furthermore, the influence of *CDC20* on cell migration and invasion was investigated in vitro. The outcomes demonstrated a significant inhibition in the migratory and invasive capacities of EJ and T24 cells upon *CDC20* downregulation (Figure 7E,F). These observations underscore the pivotal role of *CDC20* in promoting the proliferation, migration, and invasion of bladder cancer cells.

## 4. Discussion

Bladder cancer is the most prevalent malignancy affecting the urinary tract [15]. The identification of potential biomarkers specific to bladder cancer (BLCA) that can accurately predict tumor progression is essential to improve patient prognosis. The development of bladder cancer is governed by multiple genetic alterations. Despite advances in the treatment of bladder cancer, the efficacy of currently available bladder cancer treatment strategies is unsatisfactory, particularly for advanced disease, recurrent superficial carcinoma, and treatment-resistant in situ cancer [16,17]. Therefore, there is an urgent need to discover new biomarkers to accurately predict patient prognosis and to guide novel gene therapy strategies for bladder cancer.

In this study, we utilized the weighted gene co-expression network analysis (WGCNA) method to identify a cohort of 39 genes with consistent expression patterns across both the TCGA and GSE13507 databases. From this set, we further prioritized the top eight genes—*TROAP*, *TPX2*, *TOP2A*, *KIF2C*, *AURKA*, *CDC20*, *PRC1*, and *AURKB*—based on their MCC scores obtained from the CytoHubba plugin. Notably, among these genes, *CDC20* exhibited a negative correlation with prognosis, suggesting its potential as a prognostic indicator in bladder cancer. Furthermore, gene set enrichment analysis revealed *CDC20*’s involvement in facilitating the progression of bladder cancer. These findings underscore the potential significance of *CDC20* in the advancement of bladder cancer and offer insights into its underlying mechanisms.

*CDC20* is an important regulator of the cell cycle that contains seven *WD40* repeat sequences at its C-terminus, enabling protein–protein interaction [18]. Its main role is to bind to the APC (Anaphase-Promoting Complex) complex, thereby promoting its ubiquitin ligase activity and facilitating the degradation of securin and cyclin B through ubiquitination. This degradation process is essential for the proper completion of eukaryotic mitosis [19]. APCs are required to form two different E3 ubiquitin ligases with *CDC20* or *CDH1* and recognize substrates through different recognition patterns, controlling substrate specificity and the timing of degradation [20,21]. Their roles and regulatory mechanisms in the cell cycle are different. CDH1 acts mainly in late mitosis and G1 phase and remains unchanged or reduced in human cancers as a tumor suppressor [22,23,24]. In contrast, *CDC20* is frequently overexpressed in various types of cancers and is oncogenic [25,26]. For example, overexpression of *CDC20* leads to defects in the spindle assembly checkpoint (SAC) and premature disruption of Pds1/securin, resulting in aneuploidy [27]. It was found that depletion of endogenous *CDC20* in different cancer cell lines induced mitotic arrest and subsequent cell death and effectively inhibited tumor growth by *CDC20* ablation [28,29]. Conversely, deletion of the rotor component regulating inhibition of the *CDC20* checkpoint promotes tumor development [30].

To further understand the potential mechanisms of the oncogenic role of *CDC20*, we performed protein–protein interaction (PPI) network analysis of *CDC20*-interacting proteins, as well as gene set enrichment analysis (GSEA). GSEA results showed that *CDC20* highly expressed specimens were enriched in signaling pathways that are closely associated with tumor development, such as *E2F*, *TARGETS*, *G2M* checkpoint, and inflammatory response pathways. This finding suggests a correlation between *CDC20* expression and the immune microenvironment. Subsequently, we mined the differential expression and correlation between *CDC20* and immune-infiltrating cells. Our analysis revealed that *CDC20* expression was positively correlated with the number of Th2 and Th1 cells, and inversely correlated with the number of NK cells and mast cells. These findings further support the impact of *CDC20* levels on immune activity within the tumor microenvironment (TME). Notably, a significant proportion of Th2 cells is responsible for the increased infiltration of M2 macrophages and eosinophils in the TME through the secretion of IL-5 and IL-13, as well as the regulation of TGF-β secretion and immunosuppressive responses [31]. In addition, excessive activation of Th2 differentiation can profoundly impair the immune response and promote tumor growth [32]. Thus, *CDC20* appears to regulate TME by modulating the abundance of these immune cells. Further investigations are needed to elucidate the exact mechanisms by which Th cells induce pro-tumorigenic immune responses.

To validate the aforementioned findings, we initially confirmed the relatively high expression of *CDC20* in the human bladder cancer cell lines EJ and T24, while its expression was relatively low in the immortalized human bladder epithelial cell line SV-HUC-1, using quantitative polymerase chain reaction (QPCR). Subsequently, we performed *CDC20* knockdown experiments in EJ and T24 cells to assess the impact on cell proliferation, migration, and invasion abilities. Our results demonstrated that EJ and T24 cells with reduced *CDC20* expression exhibited decreased proliferation, migration, and invasion abilities. These findings are consistent with previous reports of *CDC20* overexpression in various malignancies [33].

In conclusion, we have identified potential prognostic biomarkers by WGCNA, using data from TCGA and GEO databases to confirm a significant association between *CDC20* expression and overall survival of bladder cancer patients. Available evidence suggests that *CDC20* could be a promising target for anticancer therapy [29]. The bioinformatics approach utilized in this study provides a new perspective to explore the role of *CDC20* in bladder cancer and offers valuable insights into the development of anticancer drugs as well as feasible gene therapy strategies [29,34]. However, this investigation has several limitations that warrant consideration. The retrospective nature of the data sourced from public databases (TCGA and GEO) introduces potential biases in patient selection and clinical annotations, while the absence of an independent, multi-center cohort for external validation restricts the generalizability of *CDC20*’s prognostic value. Although bioinformatics analyses implicated *CDC20* in cell cycle regulation and immune modulation, these findings lack comprehensive experimental validation, as the study relied on only two cell lines and omitted in vivo models to assess *CDC20*’s role in a complex tumor microenvironment. Furthermore, the therapeutic potential of *CDC20* remains speculative without rigorous preclinical assessment of its targeting efficacy and safety. Future work should prioritize validating these results in prospective cohorts, elucidating the underlying molecular mechanisms, and evaluating *CDC20*’s translational relevance through functional studies in diverse models.

## 5. Conclusions

In summary, through WGCNA and validation using datasets from TCGA and GEO databases, we have identified potential prognostic biomarkers, demonstrating a significant correlation between *CDC20* expression levels and overall survival in bladder cancer patients. Current evidence strongly indicates that *CDC20* represents a promising therapeutic target for anticancer treatment. Further experimental studies are needed to fully elucidate the molecular mechanisms of *CDC20* in bladder cancer progression and treatment.

## Figures and Tables

**Figure 1 diagnostics-15-03016-f001:**
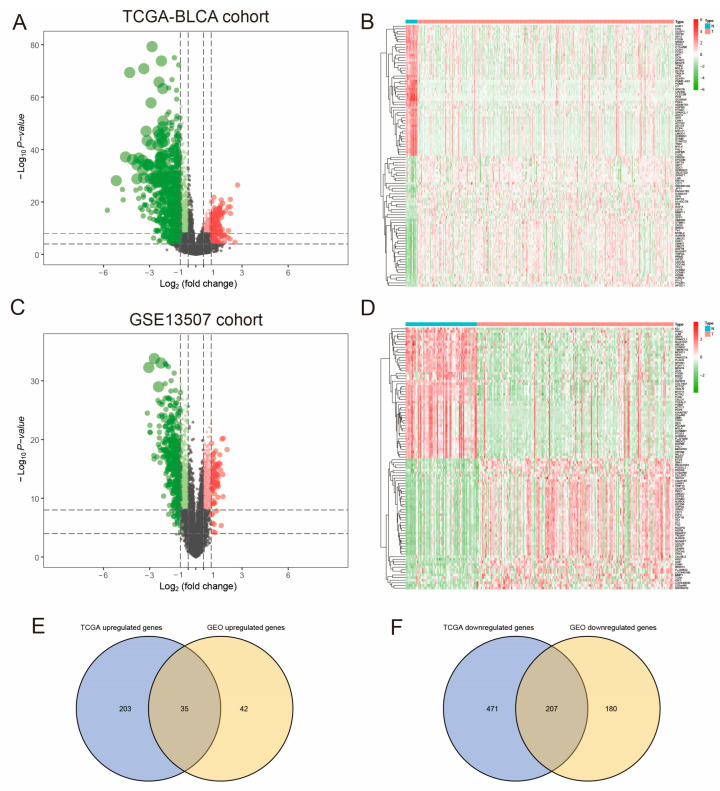
Identification of differentially expressed genes (DEGs) in bladder cancer. (**A**–**D**) Volcano plots and heatmaps displaying DEGs from the TCGA and GSE13507 datasets, respectively (cutoff criteria: |LogFC| > 1, FDR < 0.05). (**E**,**F**) Venn diagram illustrating the overlap of 35 co-upregulated and 207 co-downregulated genes common to both datasets.

**Figure 2 diagnostics-15-03016-f002:**
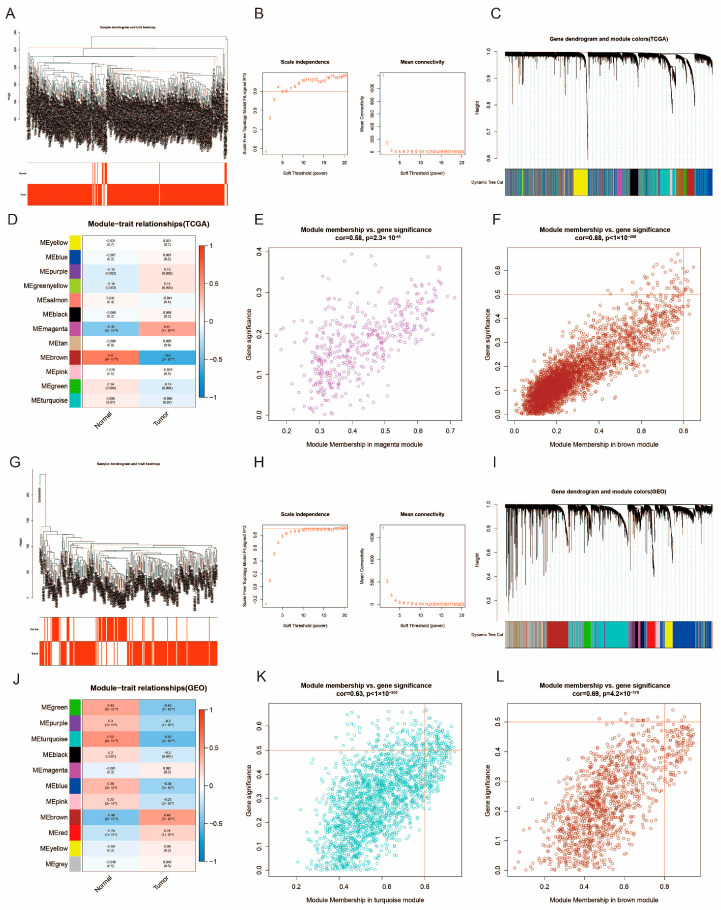
WGCNA in the TCGA-BLCA cohort and GSE13507 cohort. (**A**) Sample dendrogram and trait heatmap of the TCGA-BLCA cohort; (**B**) Scale independence and mean connectivity in the TCGA-BLCA cohort; (**C**) Gene dendrogram and modules before merging in the TCGA-BLCA cohort; (**D**) Pearson correlation analysis of merged modules and CAF score in the TCGA-BLCA cohort; (**E**) Scatterplot of MM and GS from the magenta module in the TCGA-BLCA cohort; (**F**) Scatterplot of MM and GS from the brown module in the TCGA-BLCA cohort; (**G**) Sample dendrogram and trait heatmap of the GSE13507 cohort; (**H**) Scale independence and mean connectivity in the GSE13507 cohort; (**I**) Gene dendrogram and modules before merging in the GSE13507 cohort; (**J**) Pearson correlation analysis of merged modules and CAF score in the GSE13507 cohort; (**K**) Scatterplot of MM and GS from the turquoise module in the GSE13507 cohort; (**L**) Scatterplot of MM and GS from the brown module in the GSE13507 cohort. The purple (TCGA) and brown (GSE13507) modules demonstrated the strongest positive correlations with bladder cancer.

**Figure 3 diagnostics-15-03016-f003:**
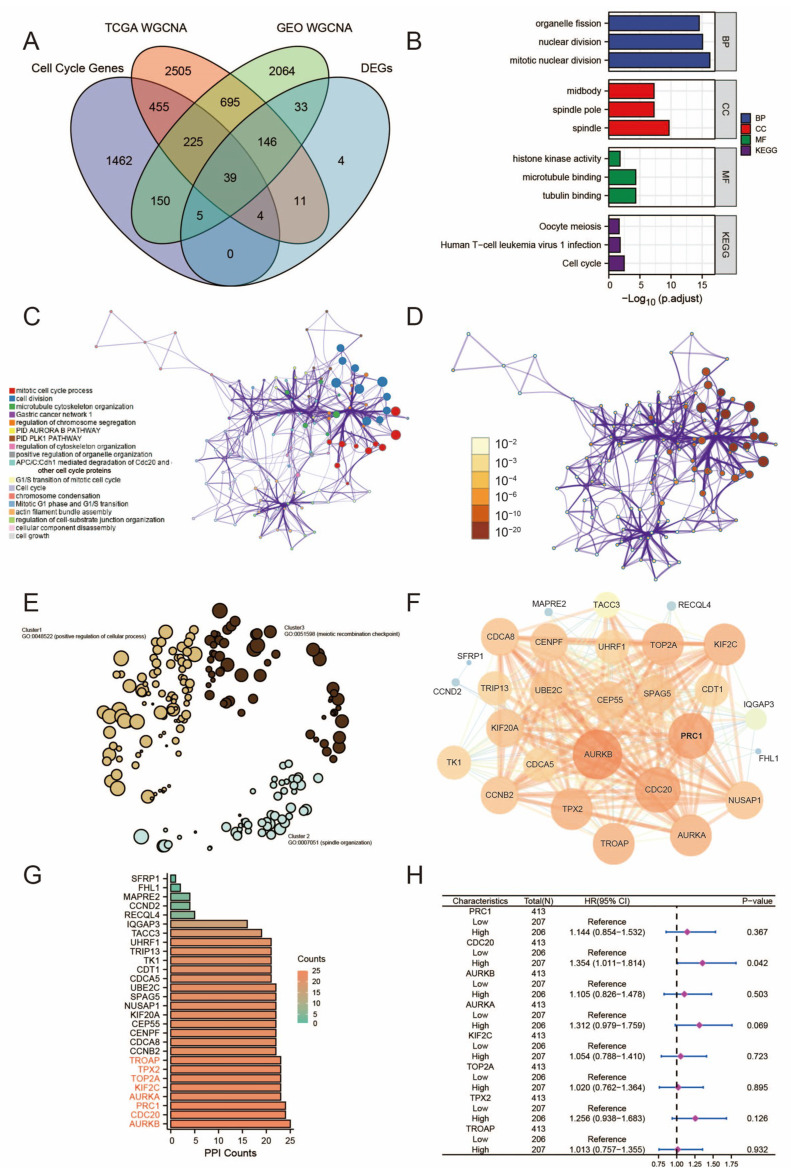
GSEA and visualization of the PPI network. (**A**) The Venn diagram of genes among the two DEG lists and the two lists of co-expression genes. A total of 39 overlapping differential co-expression genes are detected; (**B**) Functional enrichment analysis of differential co-expression genes (**C**–**E**) The GO enrichment analysis circular network plot presents a network of relationships between genes and their associated signaling pathways. The color of the node represented its cluster identity. Gene Ontology terms are significantly enriched in biological processes related to mitotic regulation and spindle organization, as well as molecular functions involving microtubule binding. KEGG pathway analysis reveals strong associations with cell cycle regulation and HTLV-1 infection pathways. (**F**) PPI network of overlapping genes constructed using the STRING database and visualized in Cytoscape. Isolated nodes were excluded from the analysis. (**G**) Hub gene identification using the Maximum Clique Centrality algorithm revealed eight central genes: *TROAP*, *TPX2*, *TOP2A*, *KIF2C*, *AURKA*, *CDC20*, *PRC1*, and *AURKB*. (**H**) Univariate Cox regression analysis of OS of hub genes in BCa. Overall survival analysis of the eight candidate hub genes was performed using the TCGA dataset. Univariable Cox regression analysis demonstrated that among all candidates, only elevated CDC20 expression showed a statistically significant correlation with reduced overall survival (*p* < 0.05), identifying it as a prognostic biomarker for bladder cancer patients.

**Figure 4 diagnostics-15-03016-f004:**
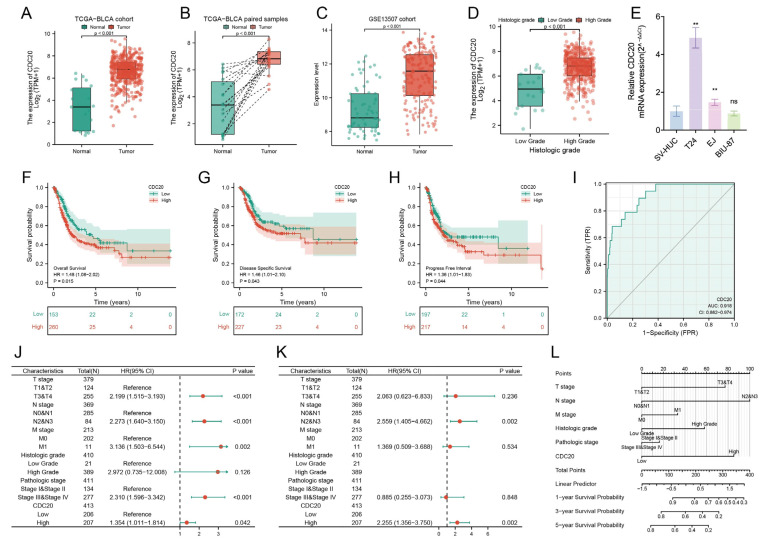
Clinical application of *CDC20* in bladder cancer. (**A**–**D**) Box plots representing *CDC20* expression level in the TCGA-BLCA and GSE 13507 dataset. The expression of *CDC20* is higher in the bladder cancer and high-grade group. (**E**) Relative expression of *CDC20* in SV-HUC, EJ, T24 and BIU-87 cells. The expression of *CDC20* is higher in the T24 and EJ cell lines. (**F**–**H**) Overall survival, disease-specific survival, and progress-free interval for *CDC20* in bladder cancer based on Kaplan–Meier plotter. High *CDC20* expression is associated with poor prognosis. (**I**) ROC curve for *CDC20* in bladder cancer patients (**J**,**K**) Multifactor Cox analysis for bladder cancer in the TCGA-BLCA and GSE 13507 dataset. (**L**) Nomogram for the diagnosis of bladder cancer. ** *p* < 0.01.

**Figure 5 diagnostics-15-03016-f005:**
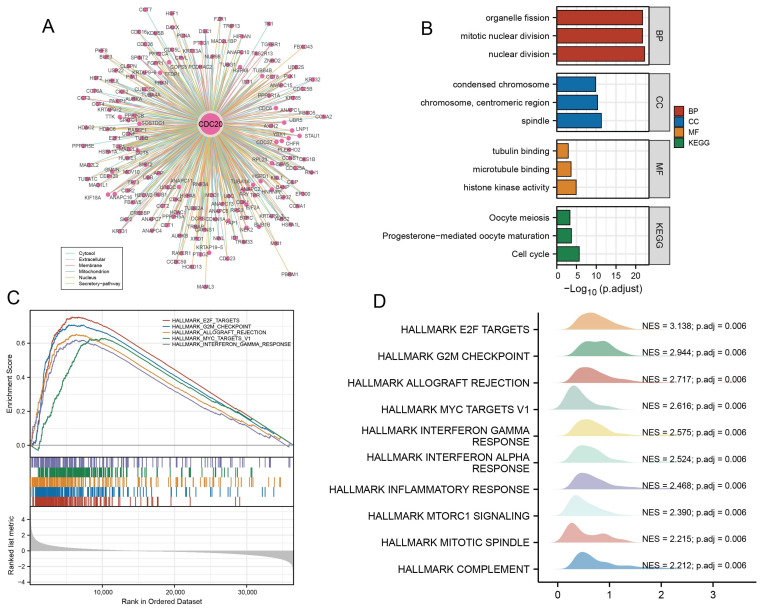
Functional enrichment of *CDC20* and validation of GSEA. (**A**) PPI network of *CDC20* with other proteins involved in bladder cancer (**B**) The GO enrichment analysis and KEGG pathway analysis of *CDC20*-associated genes. Gene Ontology (GO) enrichment analysis of *CDC20*-interacting genes reveals significant associations with mitotic regulation, chromosome organization, and microtubule binding. KEGG pathway analysis shows predominant enrichment in cell cycle regulation and progesterone-mediated oocyte maturation pathways. (**C**,**D**) Relative pathways associated with the expression of *CDC20* are shown in enrichment plots by GSEA. Gene Set Enrichment Analysis (GSEA) using Hallmark gene sets demonstrates that high *CDC20* expression is significantly enriched in E2F targets, G2/M checkpoint, inflammatory response, and immune-related pathways, suggesting its dual role in cell cycle progression and tumor microenvironment modulation.

**Figure 6 diagnostics-15-03016-f006:**
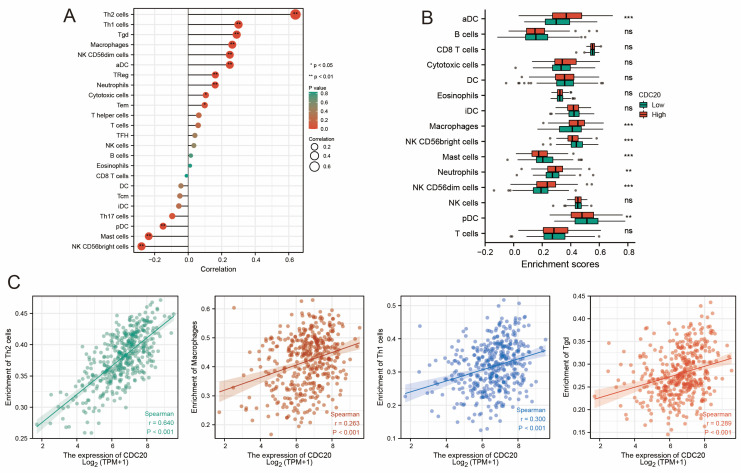
Correlation between *CDC20* expression and the immune microenvironment. (**A**) Heatmap showing Correlation between 24 kinds of TICs in TCGA-BLCA samples. (**B**) Immune infiltration analysis between the *CDC20* high- and low-expression groups. (**C**) Correlation analysis between *CDC20* and immune cells. Correlation analysis reveals *CDC20* expression is positively associated with Th1, macrophages, Tgd, and Th2 cell infiltration. * *p* < 0.05, ** *p* < 0.01, *** *p* < 0.001.

**Figure 7 diagnostics-15-03016-f007:**
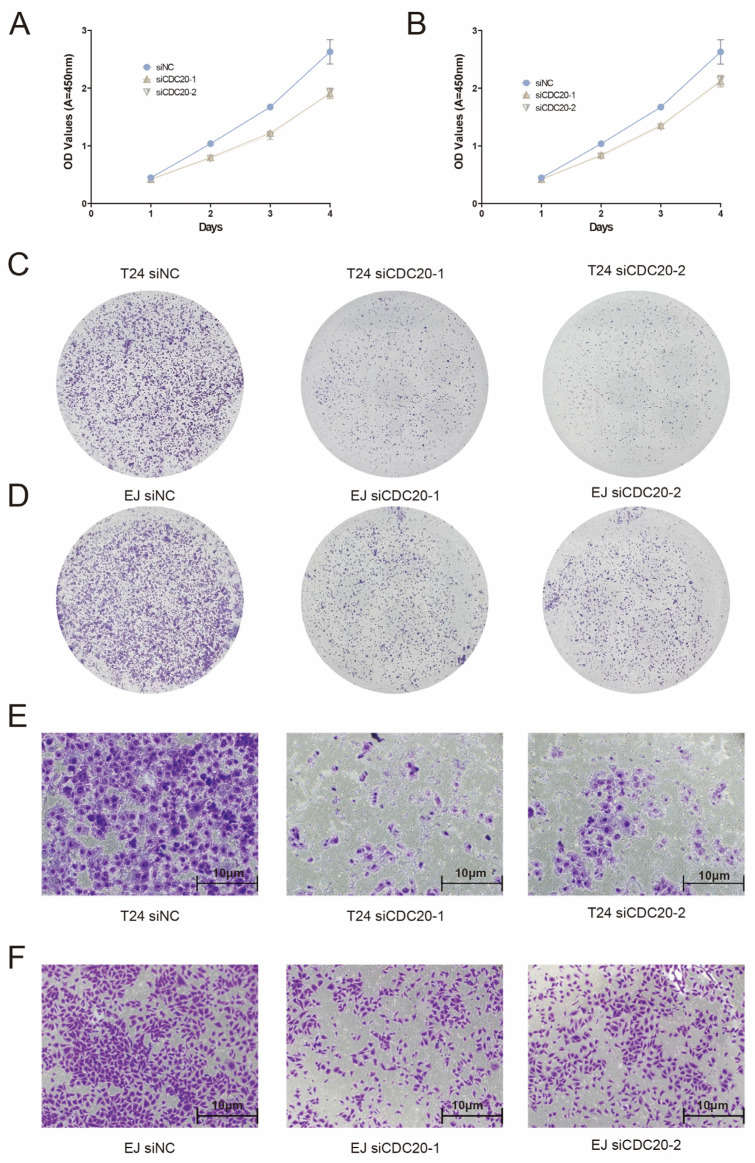
Experimental validation of *CDC20*. (**A**,**B**) CCK-8 assay demonstrating significant inhibition of cell proliferation in EJ and T24 cells following *CDC20* knockdown. (**C**,**D**) Representative images of colony formation assays showing reduced clonogenic capacity in EJ and T24 after *CDC20* depletion. (**E**,**F**) Transwell migration reveals impaired migratory and invasive capabilities in *CDC20*-deficient EJ and T24 cells. Scale bars: 10 μm. All data represent mean ± SD of three independent experiments. Original images are available in Appendix A.

**Table 1 diagnostics-15-03016-t001:** Clinical characteristics of bladder cancer patients with high and low expression of *CDC20*.

Characteristic		Low Expression of *CDC20*	High Expression of *CDC20*	*p*
Total, *n*		207	207	
Age, *n* (%)	≤70	124 (30%)	110 (26.6%)	0.197
	>70	83 (20%)	97 (23.4%)	
Gender, *n* (%)	Female	52 (12.6%)	57 (13.8%)	0.655
	Male	155 (37.4%)	150 (36.2%)	
Histologic grade, *n* (%)	High Grade	186 (45.3%)	204 (49.6%)	<0.001
	Low Grade	20 (4.9%)	1 (0.2%)	
Pathologic stage, *n* (%)	Stage I	3 (0.7%)	1 (0.2%)	0.570
	Stage II	68 (16.5%)	62 (15%)	
	Stage III	66 (16%)	76 (18.4%)	
	Stage IV	69 (16.7%)	67 (16.3%)	
T stage, *n* (%)	T1	3 (0.8%)	2 (0.5%)	0.117
	T2	59 (15.5%)	60 (15.8%)	
	T3	90 (23.7%)	106 (27.9%)	
	T4	38 (10%)	22 (5.8%)	
N stage, *n* (%)	N0	117 (31.6%)	122 (33%)	0.153
	N1	18 (4.9%)	28 (7.6%)	
	N2	46 (12.4%)	31 (8.4%)	
	N3	4 (1.1%)	4 (1.1%)	
M stage, *n* (%)	M0	117 (54.9%)	85 (39.9%)	1.000
	M1	6 (2.8%)	5 (2.3%)	
Lymph vascular invasion, *n* (%)	No	56 (19.8%)	74 (26.1%)	0.008
	Yes	91 (32.2%)	62 (21.9%)	
Primary therapy outcome, *n* (%)	PD	29 (8.1%)	41 (11.5%)	0.289
	SD	15 (4.2%)	16 (4.5%)	
	PR	12 (3.4%)	10 (2.8%)	
	CR	127 (35.6%)	107 (30%)	

**Table 2 diagnostics-15-03016-t002:** Multivariate Cox Regression Analysis of Factors Associated with Overall Survival in Bladder Cancer Patients.

Characteristics	Total (*N*)	Odds Ratio (OR)	*p* Value
T stage (T3&T4 vs. T1&T2)	380	1.000 (0.651–1.536)	1.000
N stage (N1&N2&N3 vs. N0)	370	0.889 (0.579–1.361)	0.587
M stage (M1 vs. M0)	213	1.147 (0.321–3.929)	0.825
Histologic grade (High Grade vs. Low Grade)	411	21.935 (4.505–395.580)	0.003
Pathologic stage (Stage III&Stage IV vs. Stage I&Stage II)	412	1.194 (0.790–1.806)	0.400

## Data Availability

Publicly available datasets (GSE13507 and GSE32894) were analyzed in this study. All the datasets were obtained from the GEO (http://www.ncbi.nlm.nih.gov/geo, accessed on 21 November 2025) database.

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
