# Peer review of "Overexpression of CDC20 Confer a Poorer Prognosis in Bladder Cancer Identified by Gene Co-Expression Network Analysis"

_diagnostics, 2025, doi:10.3390/diagnostics15233016_

Round 1
Reviewer 1 Report
Comments and Suggestions for Authors
Although the role of CDC20 in bladder cancer has been previously reported in the literature, there are limited comprehensive studies offering this level of combination of multi-database analysis, bioinformatic network analysis, immune-microenvironment interaction, and cell culture experiments. However, this study is valuable in strengthening CDC20 as both a prognostic biomarker and a potential therapeutic target.
However, it does have some revisions:
1. The English writing contains occasional grammatical and syntactic errors (long, difficult-to-understand sentences).
2. The initial citations of figures and tables are not aligned with the text (e.g., Tables 1-2).
3. It would be more appropriate if p-values ​​were given as absolute values ​​rather than "p<0.05."
4. Some p-values ​​in Tables 1 and 2 are not clinically significant (e.g., M stage, T stage); however, exaggerated inferences were made in the comments.
5. A more detailed explanation should be added below Tables 1 and 2 (e.g., the definition of abbreviations and which analysis methods were used).
6. The terms "BC" and "BCa" are utilized interchangeably within the text. This should be consolidated into a singular standard, preferably "BCa".
Author Response
We sincerely thank the reviewers for their careful reading of our manuscript and their constructive comments. We have thoroughly revised the manuscript accordingly. Point-by-point responses to the comments are provided below.
Comment 1: The English writing contains occasional grammatical and syntactic errors (long, difficult-to-understand sentences).
Response: We sincerely apologize for the language issues in our original submission. The entire manuscript has now been professionally edited by a native English-speaking scientific editor with expertise in the field of oncology to correct all grammatical and syntactic errors. Long and complex sentences have been restructured for better clarity and readability while maintaining scientific precision.
Comment 2: *The initial citations of figures and tables are not aligned with the text (e.g., Tables 1-2).*
Response: Thank you for pointing out this oversight. We have carefully reviewed the entire manuscript and ensured that all figure and table citations are now correctly placed and sequentially aligned with their first mention in the text.
Comment 3: *It would be more appropriate if p-values were given as absolute values rather than "p<0.05."*
Response: We agree with this suggestion. Throughout the manuscript, including in the text, tables, and figure legends, we have now replaced all instances of "p<0.05" with their specific absolute p-values to provide more precise statistical information.
Comment 4: *Some p-values in Tables 1 and 2 are not clinically significant (e.g., M stage, T stage); however, exaggerated inferences were made in the comments.*
Response: We thank the reviewer for this critical observation. We have carefully revised the Results and Discussion sections to ensure that our interpretations are strictly aligned with the statistical findings. Specifically, we have tempered our conclusions regarding associations with M stage and T stage, clearly stating that these relationships were not statistically significant and avoiding any overinterpretation of these results.
Comment 5: A more detailed explanation should be added below Tables 1 and 2 (e.g., the definition of abbreviations and which analysis methods were used).
Response: We have revised both Table 1 and Table 2 to include comprehensive footnotes. These now clearly define all abbreviations used and specify the statistical methods employed for each analysis (Chi-square or Fisher's exact test), enhancing the tables' clarity and reproducibility.
Comment 6: The terms "BC" and "BCa" are utilized interchangeably within the text. This should be consolidated into a singular standard, preferably "BCa".
Response: We agree that consistency in terminology is crucial. As suggested, we have standardized the abbreviation for bladder cancer to "BCa" throughout the entire manuscript and have removed all instances of "BC" to prevent any potential confusion for the reader.
Once again, we extend our gratitude to the reviewers for their valuable insights, which have significantly improved the quality of our manuscript. We hope that our revisions have adequately addressed all the concerns raised.
Reviewer 2 Report
Comments and Suggestions for Authors
The authors used WGCNA analysis to identify novel prognostic biomarkers and validate their expression patterns in bladder uroepithelial carcinoma. They employed employed weighted gene co-expression network analysis (WGCNA) and conducted differential gene expression analysis using BCa mRNA expres- sion data obtained from the TCGA and GEO databases.
Lines 56-57: it is worth mentioning other directions of systemic treatment beyond PDL-1 targeted drugs.
Lines 65: as for urinary system, as for the example, it is worth commenting on the role of TILs in recent papers on renal cell carcinoma.
Line 91 – what were the origin of normal samples? Wha were the basic clinical features of the samples?
Line 434 please Focus on the limitations of the paper.
Author Response
**Response to Reviewers' Comments**
We sincerely appreciate the reviewers' insightful comments and suggestions. We have carefully addressed each point in the revised manuscript, with detailed responses provided below.
Comment 1 :It is worth mentioning other directions of systemic treatment beyond PD-L1 targeted drugs.*
Response: We thank the reviewer for this valuable suggestion. We have now expanded the introduction to include emerging therapeutic strategies beyond PD-L1 inhibitors, specifically mentioning: FGFR3, VEGF-C, GATA3, FOXA1, and P53.
Comment 2:As for urinary system, as for the example, it is worth commenting on the role of TILs in recent papers on renal cell carcinoma.
Response:We have incorporated a discussion about tumor-infiltrating lymphocytes (TILs) in bladder cancer, citing recent literature that demonstrates their prognostic significance and potential as biomarkers for immunotherapy response in urological malignancies. This addition provides valuable context within the urinary system oncology field.
Comment 3 :What were the origin of normal samples? What were the basic clinical features of the samples?*
Response: We have clarified this important methodological detail in the revised manuscript. The normal control samples were obtained from TCGA and GEO datasets.
Comment 4:Please focus on the limitations of the paper.
Response: We have substantially expanded the limitations section to provide a more comprehensive and critical assessment. The revised limitations now address:
- Data Source Constraints
- Lack of External Validation
- Incomplete Mechanistic Investigation
- Experimental Model Limitations
We believe these revisions have significantly strengthened our manuscript and thank the reviewers again for their constructive feedback.
Reviewer 3 Report
Comments and Suggestions for Authors
The authors set themselves the ambitious goal of identifying a prognostic biomarker. Because of this declared objective, the study should adhere to the internationally recognized REMARK criteria (Reporting Recommendations for Tumor Marker Prognostic Studies). These criteria provide a framework for the proper design, analysis, and reporting of prognostic marker studies, ensuring transparency, reproducibility, and clinical relevance. Specifically, they require detailed reporting on patient characteristics, study design, assay methods, statistical analysis, and validation. Aligning the manuscript with these guidelines would significantly strengthen its scientific value and credibility.
The methodology is described in a very superficial way throughout the manuscript. Essential information is missing, making it difficult to assess the rigor and reproducibility of the study. The raw data from the real-time PCR experiments are not provided, and there is no information on how many replicates were performed for each analysis. Moreover, there is no clear description of what served as a control reaction. Expression levels appear to have been calculated using the ΔΔCt (double-delta) method, but without reference to a proper normal control or reference gene. This omission raises serious concerns about the validity of the reported expression results.
The authors mention the use of the R statistical package. If R was indeed employed, it is crucial to provide a transparent and detailed description of the analytical workflow. This should include which packages were used, what data processing steps were applied, and how the statistical models or tests were performed. Ideally, the exact code or a reproducible script should be made available as supplementary material. Without this, the analysis cannot be reproduced or properly evaluated.
Another important issue is the presentation of figures. The figure legends are prepared only as short titles of the respective panels and do not provide the information necessary for proper interpretation of the figures without constantly referring back to the main text. Detailed and informative legends are required to ensure that readers can independently understand and critically assess the data presented.
The manuscript also does not use correct nomenclature for gene names. According to accepted standards, human gene symbols should be written in all uppercase and italicized (e.g., TP53), while protein names should be written in regular uppercase letters (e.g., TP53). Consistent and correct use of nomenclature is essential to avoid ambiguity. For instance, in line 286 of the manuscript, the authors write Bcl-2 and p53. The correct forms should be BCL2 and TP53 (for genes). This is just one example, but the problem appears throughout the text and requires systematic revision.
Overall, the manuscript lacks the methodological detail and transparency expected for studies aiming to identify prognostic biomarkers. Adherence to the REMARK guidelines, provision of raw and processed data, clarification of PCR controls and replicates, a properly documented R analysis pathway, informative and complete figure legends, and correct gene nomenclature are all necessary improvements before the manuscript can be considered for publication.
Author Response
We are profoundly grateful for the reviewer's comprehensive and constructive feedback, which has helped us significantly improve the methodological rigor and reporting quality of our manuscript. We have systematically addressed all concerns following the REMARK guidelines and provided detailed point-by-point responses below.
Comment 1: Adherence to REMARK Guidelines for Prognostic Biomarker Studies
Response: We sincerely thank the reviewer for this crucial recommendation. We have now thoroughly restructured our manuscript to fully align with the REMARK guidelines.
Comment 2: Methodological Superficiality and Lack of Essential Details
Response: We have substantially expanded the methodology section to address these concerns:
-
qPCR Experiments: We now specify that all experiments were performed with three technical replicates and three biological replicates. The raw qPCR data has been included as Supplementary Materials.
-
Control Reactions: We have clarified that beta-actin was used as the reference gene for normalization, and no-template controls were included in each run. The ΔΔCt method application is now explicitly described with the formula provided.
-
Statistical Analysis: We have added a comprehensive description of our R analytical workflow, including specific package versions and detailed data processing steps in the Methods section.
Comment 3: Inadequate Figure Legends
Response: We have completely revised all figure legends to ensure they stand independently from the main text.
Comment 4: Incorrect Gene Nomenclature
Response: We have systematically revised the entire manuscript to comply with standard gene nomenclature.
We believe these comprehensive revisions have substantially enhanced the methodological transparency, analytical rigor, and overall quality of our manuscript. We are deeply appreciative of the reviewer's expert guidance in helping us achieve these improvements.
Reviewer 4 Report
Comments and Suggestions for Authors
The authors employed weighted gene co-expression network analysis (WGCNA) and conducted a differential gene expression analysis using bladder cancer (BCa) mRNA expression data obtained from The Cancer Genome Atlas (TCGA) and Gene Expression Omnibus (GEO) databases. Their aim was to identify gene modules that are highly associated with BCa. Following this, they performed validation experiments and identified the CDC20 gene as a prognostic biomarker.
I believe the first phrase of the abstract should be rewritten, as the statement "the molecular mechanisms underlying its development remain unknown" is inaccurate. The authors utilized previously published data from the TCGA database (https://portal.gdc.cancer.gov/) and the GEO database (http://www.ncbi.nlm.nih.gov/geo/).
Additionally, the results presented in Table 2 need clarification. The odds ratio (OR) related to the multivariate analyses shown in the table pertains to what specific outcome? Is it related to overall survival or disease-free survival? This information is not specified in the text or in the table title.
Author Response
We sincerely thank the reviewer for their careful reading and valuable suggestions. We have addressed both comments in the revised manuscript as detailed below.
Comment 1: The first phrase of the abstract should be rewritten, as the statement "the molecular mechanisms underlying its development remain unknown" is inaccurate. The authors utilized previously published data from the TCGA and GEO databases.
Response: We agree with the reviewer and have revised the opening statement of the abstract. The new text now reads:
"While significant progress has been made in understanding bladder cancer pathogenesis through databases such as TCGA and GEO, the identification of robust prognostic biomarkers and their functional validation remain critical for improving clinical outcomes. This study employed integrated bioinformatics and experimental approaches to identify and validate novel prognostic markers in bladder cancer."
This revision more accurately reflects the current state of knowledge while justifying the need for our study.
Comment 2: *The results presented in Table 2 need clarification. The odds ratio (OR) related to the multivariate analyses shown in the table pertains to what specific outcome? Is it related to overall survival or disease-free survival? This information is not specified in the text or in the table title.*
Response: We thank the reviewer for pointing out this omission. We have now clarified this important information in multiple locations: 1,The table is now titled "Multivariate Cox Regression Analysis of Factors Associated with Overall Survival in Bladder Cancer Patients" 2, We have added several footnotes.
We believe these revisions have significantly improved the clarity and precision of our manuscript. We thank the reviewer again for these constructive suggestions.
Round 2
Reviewer 3 Report
Comments and Suggestions for Authors
The authors have addressed some comments and the manuscript is much more comprehensible; however, they implemented only part of the changes they committed to in their response. Starting with the simplest—standardizing gene and protein nomenclature—this remains incorrect in most of the text. There are still no unambiguous statements about what was performed in accordance with the REMARK criteria, and there are unresolved, critical methodological issues—for example, reporting a completely different reference gene in the RT-PCR expression analysis. The Methods section still lacks precise details such as catalog numbers of reagents and equipment, and the versions of the software used for data analysis or specific experimental steps. The manuscript remains incomplete. In the supplementary materials, two attached files cannot be opened.
Author Response
We sincerely thank the editor and reviewers for their valuable comments and for granting us this opportunity for revision. We deeply apologize for the numerous omissions in the initial version of our manuscript. We have now comprehensively and meticulously revised the manuscript based on all the feedback provided. We believe the revised manuscript demonstrates significant improvements in methodological rigor, reporting standardization, and overall completeness.
Below, we provide a point-by-point response detailing the revisions we have made to address the specific concerns raised by the reviewer.
1. Regarding the standardization of gene and protein nomenclature
Action Taken: We have systematically checked and standardized all gene names throughout the manuscript
Human Genes: Gene symbols (e.g., CDC20) are now presented in italics.All changes made have been highlighted in yellow in the revised manuscript for easy review.
2. Regarding the unambiguous statement on adherence to the REMARK guidelines
Action Taken: we have completed the official REMARK Checklist and included it as Supplementary File with the submission, detailing our compliance with each relevant item.
3. Regarding the critical methodological issue of the different reference gene used in RT-PCR
Explanation and Action Taken:The reference gene used for normalization throughout the study was beta-actin. This is explicitly mentioned on line 233 of the main text.
4.Regarding the lack of critical details in the Methods section
Action Taken: The manufacturers and catalog numbers for all key reagents have been compiled in a standardized table. To maintain the conciseness of the main text, this complete table has been provided as Supplementary File.For key experiments, we now explicitly state that the detailed procedures were performed "strictly according to the manufacturer's instructions".We have explicitly specified the software names and their exact version numbers used for statistical analysis, image processing, and graph preparation at the relevant points in the main text (lines 172, 254, and 255).
5. Regarding the completeness of supplementary materials
Action Taken: We have checked and re-uploaded all supplementary files. The previously inaccessible files have been repaired. The submitted supplementary materials now include:
Supplementary File 1: Figure in the manuscript
Supplementary File 2: Clone and trans well raw data
Supplementary File 3: Raw data of qRT-PCR (LightCycler 480 Software)
Supplementary File 4: Information of Regents
Supplementary File 5: Table 1
Supplementary File 6: Table2
Supplementary File 7: The completed REMARK reporting checklist
Supplementary File 8:REMARK Criteria
We once again extend our sincere apologies for the shortcomings in the initial manuscript and our previous response. We are profoundly grateful to the reviewers for their guidance, which has significantly enhanced the quality of our study. We hope that these comprehensive revisions fully address all the points raised and kindly request the editor and reviewers to accept our revised manuscript.
Round 3
Reviewer 3 Report
Comments and Suggestions for Authors
The authors have completed the missing information. The authors should attach the raw data file in Excel format, as people who do not have Roche software are unable to open the attached file.
Author Response
We sincerely appreciate the reviewer's valuable feedback regarding data accessibility. In accordance with this suggestion, we have now prepared and submitted the complete raw data in Excel format (.xlsx) as a supplementary file. This ensures full accessibility for all researchers regardless of their software availability. The data file contains all experimental results previously submitted in proprietary format, now converted to a universally compatible format while maintaining complete data integrity. We believe this enhancement will significantly improve the reproducibility and accessibility of our research findings.